# Spatial Distribution Characteristics and Influencing Factors of Traditional Villages on the Tibetan Plateau in China

**DOI:** 10.3390/ijerph192013170

**Published:** 2022-10-13

**Authors:** Liu Jin, Zongqi Wang, Xiaohong Chen

**Affiliations:** Center of Architecture Research and Design, University of Chinese Academy of Sciences, Beijing 100190, China

**Keywords:** Tibetan Plateau, traditional village, spatial distribution, influencing factor, China

## Abstract

The Tibetan Plateau is one of the world’s most extreme habitats and one of the most challenging ecosystems on the planet. Many multi-ethnic traditional villages have developed on the plateau over its long history, and are an essential component of human settlement. It is critical to research them, and it is also significant for China’s goals to make the Tibetan Plateau a distinctive ethnic cultural preservation site and a world tourist destination. While there have been limited studies focusing on villages in the entire Tibetan Plateau area, as a result, we aim to expand the field of research on the regional study of traditional villages and make progress in research throughout the Tibetan Plateau. The question addressed in this study is what the current characteristics of the distribution of traditional villages on the Tibetan Plateau are, and we attempt to propose suggestions for the preservation of traditional villages according to the distribution characteristics. Methods such as the closest neighbor index, kernel density estimates, and spatial autocorrelation analysis are used to investigate the characteristics of the spatial distribution of traditional Chinese villages on the Tibetan Plateau, as well as regression analysis of the factors that control this distribution. The findings indicate that traditional villages are unevenly distributed over the plateau, with fewer villages in the northwest and more in the southeast, showing an agglomeration type of distribution. The village distribution on provincial and municipal is uneven with a large step difference. Tibetans make up the majority of the population in the villages, but other ethnic groups are present at the margins of the plateau. The distribution of traditional villages shows “big scattered, small gather” characteristics, and one core cluster (the Hehuang Valley area of Qinghai Province) and five high-density areas (the western Sichuan Plateau; the Three Parallel Rivers area of Yunnan Province; the Yarlung Zangbo, Nyangqu, and Lhasa rivers (YZN) area of Tibet Province; the Yushu area of Qinghai Province; and the Gannan area of Gansu and Sichuan province). The natural environment has the strongest influence on the distribution of traditional villages, followed by human impacts, especially concerning the distribution of single and multi-ethnic villages, and socioeconomic factors, which have multiple influences.

## 1. Introduction

Villages are one of the fundamental forms of human settlements [1]. They are the most prevalent and widely dispersed permanent composite type of rural settlement, and they play a crucial role in the movement of people into new areas. The traditional villages discussed in the paper are Chinese traditional villages, which are chosen jointly by the national government departments, i.e., the Ministry of Housing and Urban–Rural Development of the People’s Republic of China, the Ministry of Culture of the People’s Republic of China, the State Administration of Cultural Heritage, and so on, and they do not represent all existing villages. There is a clear definition of traditional villages: Traditional villages are exemplary examples of villages and are those that were created early in the history of settlement, have better access to traditional resources, and have certain historical, cultural, scientific, artistic, social, and economic features that need to be preserved [2]. As of 2019, a total of 6819 traditional villages in China have been identified in five batches under the leadership of the State Administration of Cultural Heritage and the Ministry of Housing and Urban–Rural Development.

The Tibetan Plateau, which is the highest, largest, and most recently formed plateau on Earth, has a significant impact on the natural environment and human activity in this region [3]. The plateau is the world’s most isolated human habitat and therefore has a significant impact on human settlement culture in this area. Over recent years, China has made clear its goal of transforming the Tibetan Plateau into a location for the preservation of Chinese ethnic and cultural traits and a world tourist destination. The Tibetan Plateau is a multi-ethnic area but is populated mainly by Tibetans. Each ethnic group has undertaken the construction of villages and buildings that best suit the local conditions. These villages serve as both a conduit for human activity and a representation of local culture. To improve our understanding of the pattern of human habitat development worldwide, it is crucial to investigate the distribution of villages on the plateau.

Since the late 20th century, traditional villages have become a topic of great academic interest, and Chinese research has produced many intersecting studies in this field. The majority of current research concentrates on village space and form [4], culture and landscape [5], and conservation and development [6]. In recent years, the spatial distribution has emerged as an important research area in addition to the aforementioned studies and requires a quantitative and multidisciplinary approach. The study scales are focused on the provincial and municipal levels, and most research results have been obtained for Hunan Province [7], Shanxi Province [8], and Guangdong Province [9] in China. In the research field of all of China’s traditional villages, the study of Tong Yuquan [10], Liu Dajun [11], Li Jiangsu [12], and Cao Yingchun [13], etc., are typical. Related research has revealed that traditional Chinese villages are concentrated in four main clusters: southern Anhui–western Zhejiang, Jin, Hebei–Henan (i.e., the central plains), and southeast Guizhou. The number of areas studied has gradually increased, although mainly within these four clusters. In a word, there are barely any studies that have considered the Tibetan Plateau as a single region in the study of traditional villages’ distribution, with the exception of a few regions, e.g., southwest China and the Bashu area. As a result, it is essential for this study to analyze the distribution of traditional villages on the Tibetan Plateau from a comprehensive view, as this will make great progress in expanding the field of research on the regional study of traditional villages.

This study considers the following themes. In Section 1, we use ArcGIS and Microsoft Excel to analyze the quantitative positioning and spatial distribution of traditional villages on the Tibetan Plateau. In Section 2, we present our statistical analysis of the relationships linking the natural environment, socioeconomic factors, and human historical factors with the distribution of traditional villages and summarize our findings. In Section 3, we investigate potential approaches to the preservation of traditional communities on the Tibetan Plateau.

## 2. Materials and Methods

### 2.1. Location

The Tibetan Plateau in western China, also known as the inland plateau of Asia, extends over the region 26°00′12″ N–39°46′50″ N, 73°18′52″ E–104°46′59″ E and covers an area of 2,582,800 km^2^ (Figure 1). Because of its significant altitude, it is also referred to as the “roof of the world” [14]. The Tibetan Plateau is a difficult place for people to live, with an average temperature below 0 °C and an average altitude of more than 4000 m. It comprises a variety of landscapes including plateaus, basins, and canyons as well as a diverse range of ecological environments. It is an important region for the development of human civilization and is the source of many of the largest rivers in East and Southeast Asia. The plateau has a long history of human habitation and is home to more than 10 ethnic groups, including Tibetans, Han, Hui, and Mongols. The entire Tibet Autonomous Region, Qinghai Province, the western part of Sichuan Province, a small part of Yunnan Province, the Xinjiang Uyghur Autonomous Region, and Gansu Province are all within the administrative area of the Tibetan Plateau, along with 37 cities and 214 counties of China.

### 2.2. Data Sources

We used two types of data. First, data for the entire Tibetan Plateau, including topography and geomorphology (e.g., altitude and slope), temperature and precipitation, rivers and water bodies, transportation and road networks, administrative boundaries, ethnic groups, and municipal gross domestic product (GDP). Most of this information was obtained from the Tibetan Plateau National Data Center of China and the official websites of the Chinese government (Table 1). Second, we used data from 2019 relating to 312 traditional villages on the Tibetan Plateau collated by various administrative departments of the People’s Republic of China (P.R.C.), including the Ministry of Housing and Urban–Rural Development for the Tibetan Plateau, the Ministry of Culture and Tourism, the State Administration of Cultural Heritage, the Ministry of Finance, the Ministry of Natural Resources, and the Ministry of Agriculture and Rural Affairs. Ethnic type is important data for a traditional village, and it is mainly determined in two ways. One way is to count the ethnic population proportion in China’s seventh census data. The official websites of the provincial, municipal, and county administrations provide access to the 7th Census data. Another way is to search the published documents for each Chinese traditional village, which includes details about the village’s location, population, photos, and other details, as well as a list of the major ethnic groups. The documents can be found on the official websites of the government or the Ministry of Housing and Urban–Rural Development of each province.

### 2.3. Methods

The mixed method prevails in interdisciplinary studies enquiring about settlement morphology at diverse scales. According to Shi Bin [15], the methods could be generally classified into two categories, namely methods for depicting the overall distribution patterns and methods for probing reasons underlying the spatial distribution. Multiple indicators show considerable suitability in quantifying the spatial characteristics of the distribution of traditional villages in varying regions. The five most commonly used methods in existing studies are the kernel density analysis, nearest neighbor index method, imbalance index analysis, geographic concentration method, and global and Anselin Moran’s I analysis.

The kernel density is widely applied to depict the agglomeration pattern of traditional villages at regional and national scales. Xu Yuqian [16] identified the high and moderate density aggregation areas of 97 traditional villages along the Yellow River in Shanxi and Shaanxi. Jiaojiao Bian [17] and Chao Wu [18] used the tool to interpret the spatial distribution patterns of 6819 traditional villages in China nationwide. Other related studies include work by Wang Peijia [19], Li Mi [20], and Guang Zhongmei [21]. The nearest neighbor index is commonly used for determining spatial distribution patterns of points [7,22]. Additionally, some studies used this method to analyze the distribution of traditional villages in the Beijing–Tianjin–Hebei region [23], Shaanxi province [24], and the Jialing River Basin [25]. Imbalance index analysis reflects the imbalanced degree of point elements in the area. The application of the imbalance index could be found in Xue Mingyue’s work in the Yellow River region [26] and Ju Xiaoxiao‘s research on Zhejiang, Anhui, Shaanxi, and Yunnan [27]. The geographic concentration method is capable of the concentration of point elements in a certain area. Liu Dajun [11] used this method to analyze the nationwide spatial distribution pattern of traditional villages in China. Moran’s I is a measure of spatial autocorrelation which is characterized by a correlation in a signal among nearby locations in space.

Many studies have integrated multiple indicators to quantify the spatial distribution of traditional villages. Based on the research above, we used three research approaches. The first approach was quantitative positioning, which uses fundamental geographic and socioeconomic information from the 312 traditional villages on the Tibetan Plateau to create the base data for this study. The data were exact and quantified for graphical analysis using ArcGIS.) The second approach was spatial overlay analysis, which aims to find the spatial distribution characteristics of traditional villages by using ArcGIS and EXCEL (Table 2). The methodology of the second approach, as shown in Figure 2, states that the kernel density is adopted to find the density distribution of villages [28]; the nearest neighbor index is used to measure the distribution type [17]; the imbalance index [29] and geographic concentration index [27] are to find the degree of imbalance and concentration of villages at provincial and city levels; Global and Anselin Moran’s I index can show the spatial autocorrelation of the villages, which is benefits to find the distribution trends and areas. Third, (to investigate the factors that influence the distribution of the traditional villages, the data layers for topography (altitude), temperature, annual precipitation, road networks, river basins, ethnic composition, and per capita GDP of the Tibetan Plateau were overlaid and analyzed alongside the traditional village data in ArcGIS, and we applied the person’s correlation analysis further to analyze the extent and mechanism of how multiple factors impact traditional village’s distribution.

## 3. Results

### 3.1. Spatial Distribution Characteristics of Villages

#### 3.1.1. Distribution on the Whole Region

##### Distribution Type

The nearest neighbor index can be used to measure the type of point elements, containing three types: uniform, random, and agglomeration (Table 2, No. 1). The distribution statistics of traditional villages on the Tibetan Plateau are as follows (Figure 2): actual nearest distance $\bar{D}$ = 1.38 km; theoretical nearest distance ${\bar{D}}_{i}$ = 4.15 km; and nearest neighbor index R = 0.33, Z = –22.98, and P = 0.00. This indicates that the distribution type of traditional villages on the Tibetan Plateau is agglomeration (Figure 3).

##### Distribution Trends and Main Areas: Cluster Area and High-Density Areas of Villages

Spatial autocorrelation is a popular method for analyzing distribution patterns within objects. Based on county location and the number of traditional villages in each county, the spatial autocorrelation of villages was identified using the “Global Moran’s I” tool in ArcGIS. With a Moran’s Index value of 0.236, a z-value of 2.84 (above the standard deviation of 2.58), and a *p*-value of 0.004, the results showed that the distribution of villages was not randomly distributed but rather had a positive spatial autocorrelation. In addition, we apply the “Anselin Local Moran’s I” in ArcGIS to find significant cluster areas. As shown in Figure 4a, two High–High clusters, which means the counties with several villages are gathered here, and are located in the Hehuang Valley area, so that the Hehuang Valley area can be identified as a cluster area.

As the results above show, no significant clusters have developed inside the villages, with the exception of the Hehuang Valley area, which is tied to the Tibetan Plateau’s vast surface area. In this case, density distribution could be used to further analyze the distribution trend.

We analyzed the density distribution of traditional villages on the Tibetan Plateau using kernel density (Figure 4b). The distribution is uneven, with fewer villages in the northwest and more in the southeast. Within this distribution, there is a pattern of “large dispersion, small aggregation”, which means traditional villages are widely dispersed overall, but in some places, they show aggregation. We identified six high-density areas, comprising one highest-density area (the Hehuang valley area) and five general high-density areas (the western Sichuan Plateau area, the Three Parallel Rivers area, the Yarlung Zangbo, Nyangqu, and Lhasa rivers (YZN) area, the Yushu area; and the Gannan area).

(1)Cluster area: Hehuang Valley area

The fertile Hehuang Valley area in western Qinghai Province, which is bounded by the Yellow River and Huangshui River, is a prominent village’s core cluster area on the Tibetan Plateau, indicating it is the highest density area as well as the cluster area (Figure 4a). The 102 traditional villages in this region make up 31.38% of all the villages on the Tibetan Plateau, despite its small size (about 30,000 km^2^, or 1.1% of the overall Tibetan Plateau). Traditional villages are found mostly in the southern ethnic minority communities and around Xining. The villages are distributed unevenly across the counties (Figure 4b), with Tongren County and Xunhua Salar Autonomous County having the highest numbers of villages.

(2)High-density areas: western Sichuan Plateau, Three Parallel Rivers, YZN, Yushu, and Gannan areas

The traditional villages occur in two main high-density areas in the western Sichuan Plateau and the Three Parallel Rivers areas, which are larger in size and greater in number than the other high-density areas (Figure 4b). The western Sichuan Plateau is located in the eastern part of the Tibetan Plateau, includes the western part of Sichuan Province, and is an area of typical Kangba–Tibet culture. There are 81 traditional villages in total, making up about 25% of all villages, with Li County being the sub-group center and Danba County the secondary center. The Three Parallel Rivers area, which includes the northwest of Yunnan Province and the southern part of Sichuan Province, is situated in the southern part of the Tibetan Plateau and is a typical multi-ethnic area. There are 58 traditional villages, which make up 17.96% of the total villages. The traditional villages are uniformly scattered, with no discernible group core.

The YZN, Yushu, and Gannan areas contain mainly Tibetan villages that are relatively small-scale and independent of each other (Figure 4b). The Yarlung Tsangpo, Nyang Chu, and Kyi rivers meet in the YZN area, which is the core area for Tibetan culture and contains many ancient temples and palaces. There are 19 traditional villages in the area, making up 6.10% of all communities on the plateau. The area is small, and the villages are distributed mainly along the river in an east–west orientation. The Yushu area is located southwest of Qinghai Province, close to the center of the Tibetan Plateau. The area contains 20 villages (6.41% of the total) and is a typical Anduo–Tibetan cultural area. Yushu City is located in the center of the area. The Yushu area is small, and the villages have a concentrated distribution. The Gannan area, which makes up the majority of Gannan Prefecture in Gansu Province, is situated on the eastern edge of the Tibetan Plateau and contains 16 villages (5.13% of the total). The villages define an area that trends north-south and they have a relatively even distribution.

#### 3.1.2. Distribution at the Provincial Level

The imbalance index reflects the degree of objects’ coverage of the distribution within different areas, which is commonly related to quantity. [23], and it is used in this study to assess the balance of traditional village distribution throughout the province. According to Model 3 in Table 2, we calculated the imbalance index to be *S* = 0.60 (0 < *S* < 1). Assuming that the number of traditional villages is balanced, the imbalance index *S*′ is calculated to be 0 in the ideal state [30]. The actual calculated *S* is greater than *S*′, indicating that the distribution of traditional villages on the Tibetan Plateau is unbalanced. The provincial distribution of villages is uneven, and this is also intuitively apparent in that the Lorentz curve is more curved than the average (Figure 5), and leans slightly towards an even distribution.

According to the number of villages within the provincial allocation, we find there are three classes. The top two provinces in terms of the number of villages are Sichuan and Qinghai, together accounting for 77.24% of the total number of villages (Table 3). The second class includes the Tibet Autonomous Region and Yunnan Province, which have the same number of villages, and accounted for 18.29% of the total number of villages. The third class comprises Gansu Province and the Xinjiang Uygur Autonomous Region, which contain 4.49% of all villages.

The Tibetan Plateau occupies 26.9% of China’s total land area but contains only 4.57% of the total of 6819 traditional Chinese villages. The imbalance index for all traditional villages in China is *S* = 0.59 [17], which is similar to that for the Tibetan Plateau, and shows that the distribution balance of traditional villages is similar for both China and the plateau.

#### 3.1.3. Distribution at the City Level

Geographic concentration index is a method to measure the degree of spatial distribution and agglomeration of objects [31]. There are 37 cities on the Tibetan Plateau, 22 of which contain traditional villages. According to Model 4 in Table 2, the average geographical concentration index ($\bar{G}$) is 14.18, and we calculated the spatial concentration index of traditional villages at the city level (Table 4). The result shows the agglomeration degree of villages at the city level differs significantly (Figure 6). Ganzi Autonomous Prefecture in Sichuan Province has the most villages and the highest G value, followed by Haidong in Qinghai Province and Aba Tibetan and Qiang Autonomous Prefectures in Sichuan Province. All three have G values greater than $\bar{G}$, making up 52% of the entire area. The G value of the Huangnan Tibetan Autonomous Prefecture in Qinghai Province is close to $\bar{G}$, and the villages are relatively concentrated. The remainder of the cities on the western Tibetan Plateau and in western Qinghai Province have below-average G values, and more than half of the urban areas (12) have a G value below 5.00, reflecting the sparse distribution and a small number of villages.

#### 3.1.4. Ethnic Distribution

The Tibetan Plateau has a diverse ethnic population, and the distribution and growth of these groups have a significant impact on the creation and development of communities. Through statistics and visualization of each traditional village (Figure 7), the ethnic distribution characteristics of the villages can be found as followings. The Tibetan Plateau has more than 14 different ethnic groups, and the traditional villages include both those inhabited primarily by a single ethnic group and those inhabited by two or more ethnic groups (Table 5). The majority of the traditional villages are Tibetan, accounting for 220 villages in total (i.e., 70.51%), and are widely dispersed throughout the central part of the Tibetan Plateau (Figure 7). There are also several multi-ethnic mixed villages, which make up 8.65% of the total, and their ethnic makeup consists mainly of a mix of Tibetan and other ethnic groups, such as Tibetan and Han, Tibetan and Hui, Tibetan and Yi, Tu and Han, Menba and Luoba, and Sala and Hui. These villages are located mainly in the Three Parallel Rivers area of Yunnan Province and the Hehuang Valley area of Qinghai Province. The former mainly includes the Hui, Tibetan, Han, and Mongolian people, whereas the latter is a conglomeration of ethnic minorities, including the Tibetan, Nakhi, Nu, Mosuo, Lisu, Monpa, Lhoba, and Yi people.

### 3.2. Factors Influencing the Spatial Distribution of Traditional Villages

#### 3.2.1. Environmental Factors

##### Topography and Altitude

The Tibetan Plateau is located on the first terrain step of China and has an average altitude of over 4000 m, being higher to the northwest and lower to the southeast (Figure 8). In the northwest, the topography and alpine climate limit the number of residents in the region and they are mostly nomadic, with few villages. As the altitude decreases to the southeast, the number of villages increases.

There are 109 traditional villages at altitudes between 2500 and 3000 m (34.94% of all villages), among which 58 traditional villages are located in the Hehuang Valley area of Qinghai Province, the lowest point of the Tibetan Plateau. Only 11 villages are located at altitudes above 4000 m, all in the Tibetan area (Figure 9a). A total of 167 traditional villages occur on flat areas with a slope of <6° (53.53% of all villages), including 88 in the Hehuang Valley area and 44 in the western Sichuan Plateau Area. In areas where the slope exceeds 10°, the number of traditional villages shows a significant negative correlation with slope. Only 15 traditional villages are located in areas with a slope of >20°, mainly in the western Sichuan Plateau area where resources are relatively scarce (Figure 9b).

The above analysis shows that the lower and flatter topography of the Hehuang Valley area in Qinghai has acted as a focal point for the development of traditional villages. Furthermore, the traditional villages in the general cluster areas, such as the western Sichuan Plateau area and the Three Parallel Rivers area, occur mainly in areas with favorable topography.

##### Temperature and Precipitation

The Tibetan Plateau is a vast area with great climatic variation (Figure 10). With a reference to the Köppen climate classification, the predominating climate zone on the Tibetan Plateau is the tropical and subtropical desert and steppe where very few traditional villages are distributed for the arid harshness, while most traditional villages are located in climate zones of humid continental and continental subarctic climates where relatively abundant water resources are essential for the emergence and development of villages. There are 32 traditional villages in areas where the average annual temperature is <4 °C, with 10 of these in the Yushu area. Seventy traditional villages (22.44% of the total) are located in areas where the average annual temperature is 4–6 °C. These villages are located across the Hehuang Valley area, the Gannan area, the western Sichuan Plateau area, the Three Parallel Rivers area, and the YZN area. According to the result of Pearson’s correlation analysis (0.113), the association between the annual average temperature and the number of traditional villages shows no considerable significance. But the number of traditional villages shows a significant negative correlation with temperature as regions with higher temperatures shrink drastically (Figure 11a). Areas with average annual precipitation in the range of 300–800 mm contain 271 villages (86.86% of the total). Most of these villages are in areas with average annual precipitation of 300–400 mm (67 villages, 21.47% of the total), mainly in the Hehuang Valley area and the YZN area. Only four traditional villages are located in areas with average annual precipitation of <300 mm, mainly in the northwest of the Tibetan Plateau and other arid areas. The overall precipitation on the Tibetan Plateau is low, and 31 traditional villages are located in areas with average annual precipitation of >800 mm; these villages are concentrated in a small number of areas such as the Western Sichuan Plateau and the Three Parallel Rivers area (Figure 10b). Pearson’s correlation analysis shows the result of 0.072, which reveals that the precipitation makes no obvious positive or negative impact on traditional villages.

##### River Systems

“Water orientation” is an important principle in the selection of traditional village sites on the Tibetan Plateau (Figure 12), as water is an essential resource for living and food production, and provides a natural defensive barrier for settlements. Figure 12b shows that 187 traditional villages (59.94% of the total) are located within 10 km of the buffer zone of the water system. This feature is significant in the YZN area, the Three Parallel Rivers area, the western Sichuan Plateau area, and the Hehuang Valley area. The wide valleys in the YZN and Hehuang Valley areas contain the most developed agricultural land on the Tibetan Plateau, and the traditional villages are densely concentrated and present a piecewise group distribution. The number of traditional villages embodies a declining tendency as the distance to water resources increases. The western Sichuan Plateau area and the Three Parallel Rivers area are characterized by high mountains and deep valleys, and traditional villages are mostly distributed linearly along the valleys.

#### 3.2.2. Socioeconomic Factors

##### Transportation

The locations of roads and railways are related to the distribution of traditional villages. The road transportation system has the greatest influence on the distribution of traditional villages, whereas the railway system has little effect. County roads have the greatest influence on the spatial distribution of traditional villages and play the role of “capillaries” within the overall transportation system. The study applied Pearson’s correlation analysis to justify the association between the distribution of traditional villages and transportation. The result of the correlation coefficients between the number of traditional villages and the density of railways (−0.048), county roads (0.104), provincial roads (−0.057), and national roads (−0.045) proves that the country road system has a rather weak impact on villages. The density of county roads is highest in the Hehuang Valley area, the western Sichuan Plateau area, the YZN area, and the Yushu area, and 280 traditional villages are located in these areas (89.74% of the total; Figure 13a). More than half of the traditional villages are located within 10 km of provincial and county roads (Figure 13b).

##### Regional Gross Domestic Product

A comparative analysis of the relationship between the regional gross domestic product (GDP) and the distribution of traditional villages around 22 cities and states across the Tibetan Plateau region in 2021 (Figure 14) revealed clear stage differences in the impact of economic development on the distribution of traditional villages. The economic factor functioned obviously in two polarized conditions. In areas where the overall level of economic development is low, urbanization has a smaller impact on traditional lifestyles, which is beneficial to the preservation of traditional villages. In areas where the overall level of economic development has improved (especially with the rapid development of tourism), such as Haidong City, Aba Tibetan, Qiang Autonomous Prefecture, and Ganzi Tibetan Autonomous Prefecture, the spillover effect of economic development has promoted the protection of traditional villages, thus forming a cluster of traditional villages in these regions. According to the result of Pearson’s correlation analysis, the association coefficient between the county GDP and the number of traditional villages is −0.128. This reveals that the negative effect of high-level economic development on traditional villages tends to prevail.

#### 3.2.3. Historical Factors

The Tibetan Plateau has a long history of settlement, and the historical background of ethnic development varies greatly from region to region. The history of human activity in Tibet dates back to the Paleolithic period, 50,000 years ago [32]. In seventh century AD, the Tubo Dynasty conquered the surrounding tribes of Togukhun, Suvi, and Yangtong, and established the Tibetan nation “Bo” with common territory, language, economy, and customs, and farming and settlements developed in the Yalong River basin [33]. Since the Yuan dynasty’s reunification and the development of the Ming and Qing dynasties, numerous villages have grown in Lhasa on both banks of the river, is now the basis of this area has become a traditional village gathering area [34]; Qinghai was ruled subsequently by Xirong, Qiang, and Tubo in history. Nurtured by the Yellow River and the Huangshui River, the Hehuang Valley area is gifted with the best-cultivated land on the Qinghai-Tibetan Plateau and thus fostered an agricultural civilization. Qinghai has been under the administration of the central government since the Han dynasty and the long-lasting military-cultivation practices turned the Hehuang Valley into the core cluster area of traditional villages. The traditional villages in the region were built mostly during the Ming and Qing dynasties, and some can be traced back to before the Yuan dynasty [35]. The formation of multi-ethnic distribution patterns in Yunnan–Tibet, Sichuan–Tibet, Gansu–Qingdao, and other junction areas was accompanied by the flow of human and material resources along ancient trade routes such as the Tangfan and Tea Horse routes [36].

#### 3.2.4. The Associations and Distinctiveness among Multiple Factors

The distribution of traditional villages on the Tibet Plateau is the joint result of multiple factors. The natural settings of the plateau determined the geographical context of traditional villages. Among the diverse environmental factors, temperature plays the most significant role in shaping the distribution pattern of villages. The close associations between altitude, temperature, and precipitation fundamentally depict the trajectory of village emergences and developments. This study further explored the socioeconomic and ethnic factors which influence traditional villages in an interactive way distinctive from the decisive role played by the natural environment. The interrelated GDP and transportation infrastructure data could precisely depict the unbalanced economic development of the whole region to some extent. Despite the alleged dual impact of economic development, the statistics reveal that the negative effect of high-level economic development on traditional villages tends to prevail, and this shed some light on the wide discussion on rural tourism and the preservation of traditional villages. The above factors shape not only settlements but also ethnic patterns. This study initiated the work of associating the panoramic ethnic features with the distribution of traditional villages on the plateau. The interactive influence of ethnic factors on traditional villages played in a more complex and intangible way.

## 4. Discussion

### 4.1. How to Protect the Traditional Villages on the Tibetan Plateau Due to the Distribution Characteristics

Traditional villages on the Tibetan Plateau are a priceless and irreplaceable legacy that reflects the lives of the people who survive in this harsh environment. Over recent years, rapid economic growth and the impact of tourism have had detrimental effects on the preservation of traditional villages. It is an urgent requirement that these villages be preserved and repurposed and that their regional characteristics be investigated in light of their spatial form and distribution. Based on a comprehensive analysis of the spatial distribution characteristics and influencing factors of traditional villages, we propose the following scientific suggestions for sustainable development, helping to achieve the goal of building the Tibetan Plateau into a cultural preservation region and a world tourism destination:(1)Improve the coordinated regional development and promoting a balance in the number of selected traditional villages in different regions. China’s traditional villages, which are a context selected by the government, are closely linked to the regional economy, transportation, and tourism. Original traditional villages are easier to discover and protect in areas with convenient transportation and a good regional economy, whereas they are weaker in remote areas with poor transportation accessibility. To avoid the occurrence, the government should assume a major role in promoting more balanced and coordinated development. In China, some policies have been centered on this suggestion, but implementation has mainly taken place at the county level, where the concentrated conservation in patches is one important policy. This policy seeks to identify counties with a high concentration of traditional villages and exceptional local cultures. Following their selection, the counties will receive a fixed subsidy from the central government of China in the range of 30 to 50 million yuan. In March 2022, the Xunhua County of Qinghai province, which is part of the core cluster area of the Huanghuang Valley, was chosen for the first batch of the list. We recommend strengthening regional coordination by concentrating on the six distribution areas. A clustering effect and spillover effect should be unleashed in the main high-density distribution area—the Huanghuang Valley—to improve the quality of village conservation and encourage the development of the surrounding areas. In areas where the terrain is challenging and the number of villages is smaller, such as the western Sichuan Plateau and the Three Rivers, attention should be made to the rediscovery of traditional villages, as well as the creation of transportation and tourism infrastructure.(2)Strengthen the differentiated protection approach in accordance with the regional approach. The Tibetan Plateau’s most distinguishing features are its ethnic diversity and geographical complexity, where different ethnic groups have formed their settlement areas and, in a broad sense, cultural subdivisions through continuous adaptation to the environment over a long period. The expansion of international tourism is changing the relationship between ethnic groups and states around the globe [37], which is happening on the Tibetan Plateau. In this case, the identifiability of traditional villages dwindles, necessitating regional and typological village protection. We emphasize the employment of a zonal differentiated conservation strategy based on six high-density distribution zones, which emphasizes the preservation of the diverse regional cultures’ uniqueness.(3)Establish a system for the identification and preservation of ethnic villages on the Tibetan Plateau. The plateau has significant ethnic diversity, which is reflected in its traditional villages, and this study has described single-ethnicity settlements dominated by Tibetans, as well as mixed multi-ethnic villages. Given that China’s policy calls for the creation of a distinct ethnic reserve on the Tibetan Plateau, these distributional characteristics strongly promote the creation of an ethnic settlement identification system. The creation of this system can be approached from two angles. The first step is to distinguish between the various ethnic settlements. This can be determined by analyzing and generalizing the village’s plan, morphology, traditional architecture, and other elements. Second, specific conservation and development strategies and suggestions are put forth for various ethnic settlements based on the identification of ethnic settlement types in the earlier stage. These two methods can be used to preserve traditional settlements on the Tibetan Plateau in their entirety. These two actions will allow for the general protection of traditional Tibetan plateau villages.

### 4.2. Limitations of This Study

Although the analysis of the distribution and influencing factors of traditional villages on the Tibetan Plateau contributed to extending the study field, some limitations should be addressed in the future. First, we adopted geospatial analysis methods which are suitable for the Tibetan Plateau at a regional level, however, due to the wide area and strict geographical conditions of the plateau, the study of traditional villages is more difficult than in the plains, and related research is still in its early stages. As technology develops and village data are updated, new research methods, in particular, the multiple factor overlay analysis methods, can be further explored, enabling the results of this study to be supported or modified. Second, this study focuses on the macro-distribution of villages, for which the form, structure, landscape, and architecture of the traditional village are not discussed in this study. In future research, along with the above, the sections dealing primarily with architectural disciplines require further development.

## 5. Conclusions

This study used geographical spatial analysis to assess the properties and distribution of traditional villages to protect and identify the national culture of the Tibetan Plateau and to facilitate the protection and development of the villages. The main innovative points and conclusions are as follows:(1)The early comprehensive study of the overall Tibetan Plateau area is the key advance of this study since it expands the study field of traditional villages at the regional level. In comparison to other countries, China has several traditional villages—6819 of them are listed on the assessment list. The trend is to investigate the macro-geospatial distribution of villages as village data collection progresses. With the advancement of government-guided village data collection, a macro-level geographic study of Chinese villages has emerged as a major trend. There have been limited studies focusing on villages in the entire Tibetan Plateau area, and the study results related to the plateau only cover some parts of it, such as Qinghai Province and Sichuan Province. Furthermore, we investigated the ethnic distribution characteristics of villages, which are rarely covered in previous research but are an essential feature of the Tibetan Plateau. Consequently, we focused on the Tibetan Plateau area, applying geographical analysis to investigate the spatial distribution characteristics of traditional villages and their influencing factors and propose suggestions for the development of traditional villages in the region, in the hope of drawing more attention to it.(2)Our findings indicate that traditional villages are distributed unevenly across the Tibetan Plateau, being sparse in the northwest and abundant in the southeast, and the type of distribution is agglomeration (nearest neighbor index R = 0.33). The provincial and municipal distribution is uneven, with a large step difference. Tibetans make up the majority of the interethnic population in the villages, with other ethnic groups being mixed in marginal areas. The distribution follows a “big scattered, small gather” pattern, i.e., it is generally distributed with localized areas characterized by aggregation (Global Moran’s I Index value = 0.236, z value = 2.84). We identified one cluster (i.e., the Hehuang Valley area of Qinghai Province) and five high-density areas (i.e., the western Sichuan Plateau, the Three Parallel Rivers area of Yunnan Province, the YZN area of Tibet Province, the Yushu area of Qinghai Province, and the Gannan area).(3)The natural environment has the greatest influence on the distribution of traditional villages, followed by human history, especially concerning the distribution of single and multi-ethnic villages, and socioeconomic factors have multiple influences. The natural setting of the formation and growth of villages includes altitude, slope, temperature, and precipitation. Human history is the major determinant of the pattern of village distribution, and the core cluster area is historically a site of agricultural development. Multi-ethnic cultural mingling has influenced the distribution of villages and has also produced distinctive regional characteristics. Socioeconomic factors, including the level of economic development and the construction of road infrastructure, have influenced the layout of traditional villages. In summary, the long-term interaction of these factors has led to the present-day spatial distribution of traditional villages on the Tibetan Plateau. We have investigated the distribution of traditional villages on the sparsely populated Tibetan Plateau and the factors that control the distribution. We aim to focus on the conversion and development of traditional villages in future work.

China has proposed many strategic directions for rural revitalization and tourism development, including the building of attractive villages and small towns with distinctive features. In addition, the Tibetan Plateau is a key area for China’s plans for tourism development and cultural preservation, and is a significant element of the “Belt and Road Initiative” policy, the establishment of a national cultural reserve with Chinese characteristics, and the development of international tourist hotspots, all of which present fresh opportunities for the preservation and use of the region’s traditional villages.

## Figures and Tables

**Figure 1 ijerph-19-13170-f001:**
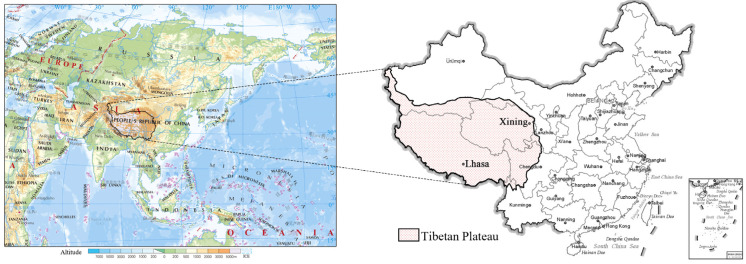
Location of the Tibetan Plateau.

**Figure 2 ijerph-19-13170-f002:**
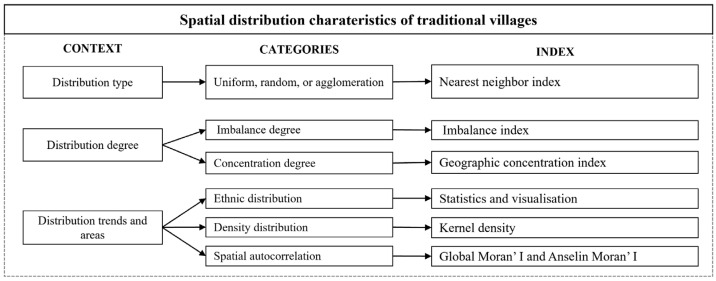
Methodology presentation on the spatial distribution characteristics of traditional villages.

**Figure 3 ijerph-19-13170-f003:**
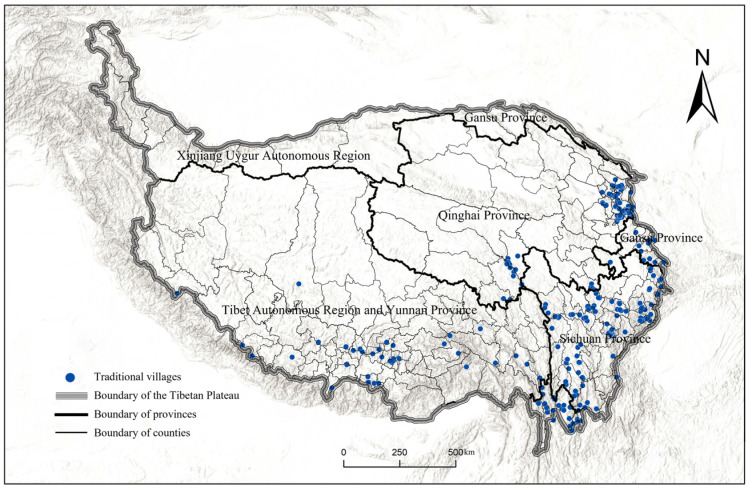
Distribution of traditional villages on the Tibetan Plateau.

**Figure 4 ijerph-19-13170-f004:**
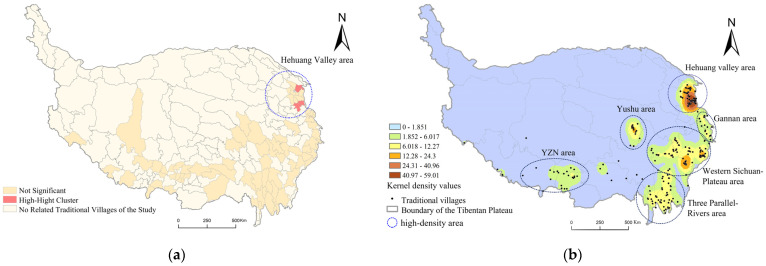
High-density and cluster areas of traditional villages on the Tibetan Plateau: (**a**) cluster map of villages based on the tool of Anselin local Moran’s I; (**b**) density map of villages based on the tool of kernel density.

**Figure 5 ijerph-19-13170-f005:**
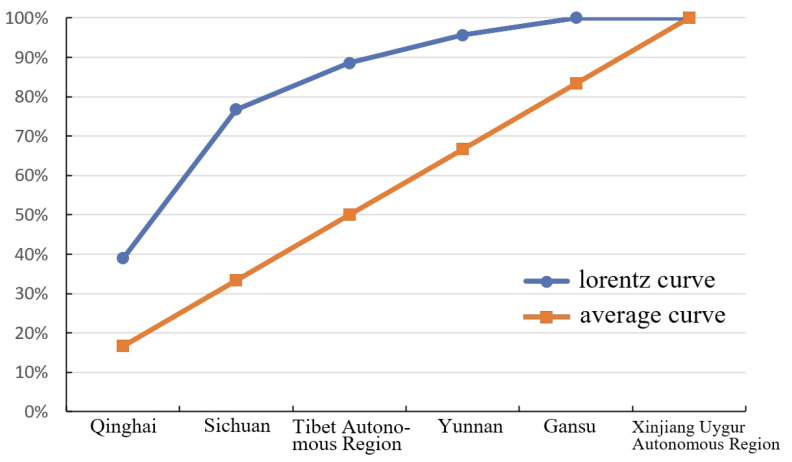
The Lorentz curve.

**Figure 6 ijerph-19-13170-f006:**
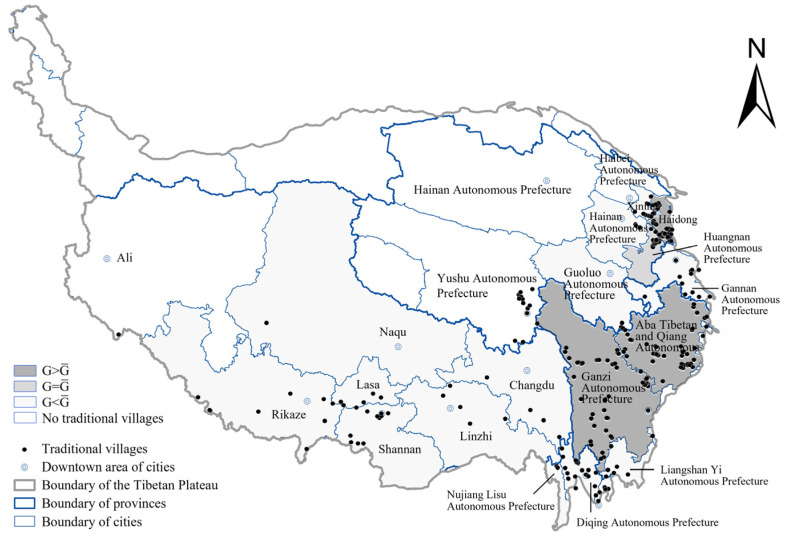
Distribution of villages at the city level on the Tibetan Plateau.

**Figure 7 ijerph-19-13170-f007:**
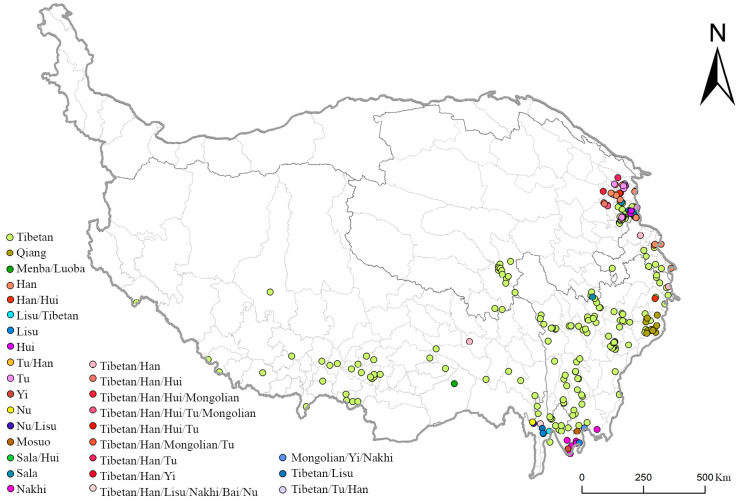
Distribution of traditional villages with different ethnic make-ups on the Tibetan Plateau.

**Figure 8 ijerph-19-13170-f008:**
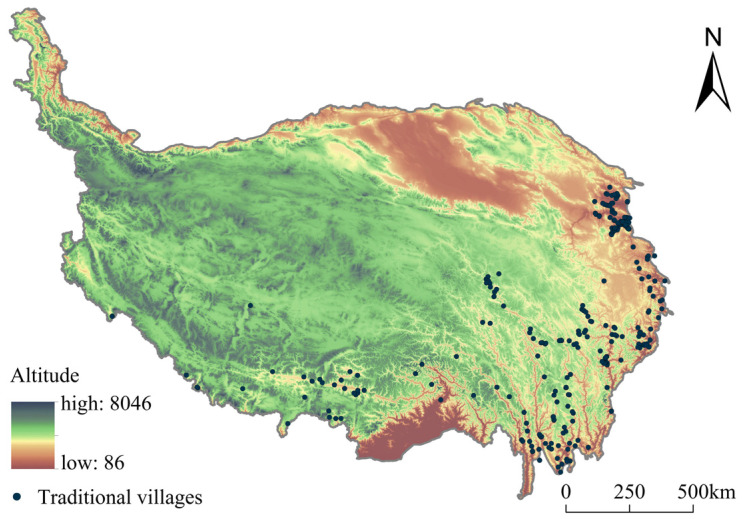
Altitudes of traditional villages coupled with altitude on the Tibetan Plateau.

**Figure 9 ijerph-19-13170-f009:**
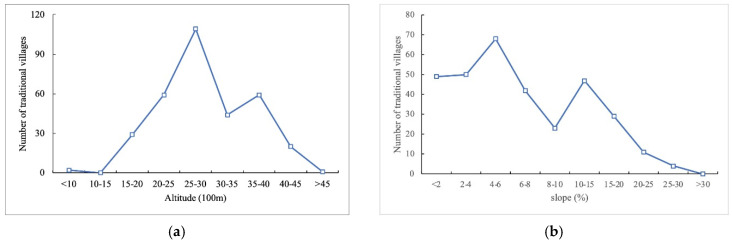
Distribution of traditional villages in terms of altitude and slope on the Tibetan Plateau: (**a**) distribution of villages by altitude; (**b**) distribution of villages by slope angle.

**Figure 10 ijerph-19-13170-f010:**
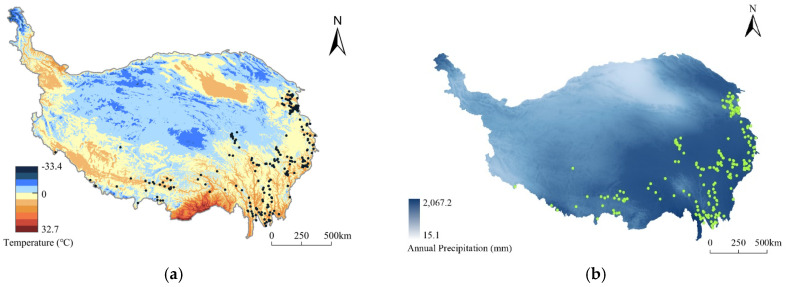
Distribution of traditional villages coupled with different conditions on the Tibetan Plateau: (**a**) the spatial overlay analysis of village distribution and annual temperature; (**b**) the spatial overlay analysis of village distribution and annual precipitation.

**Figure 11 ijerph-19-13170-f011:**
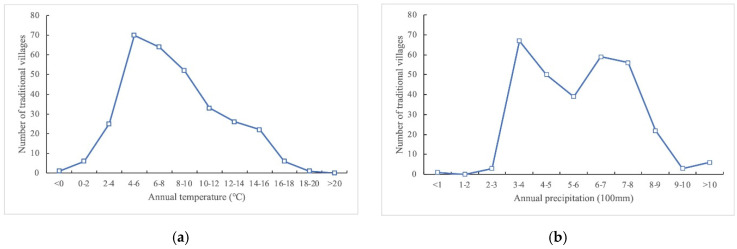
Statistics of traditional villages on the Tibetan Plateau in different annual conditions: (**a**) temperature; (**b**) precipitation.

**Figure 12 ijerph-19-13170-f012:**
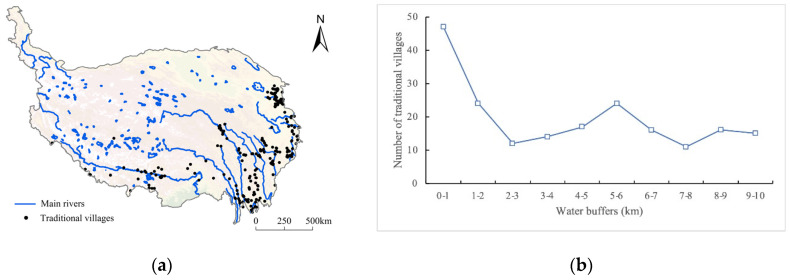
Distribution of traditional villages coupled with water systems on the Tibetan Plateau: (**a**) the spatial overlay analysis of village distribution and water systems; (**b**) statistics of traditional villages in different water buffers.

**Figure 13 ijerph-19-13170-f013:**
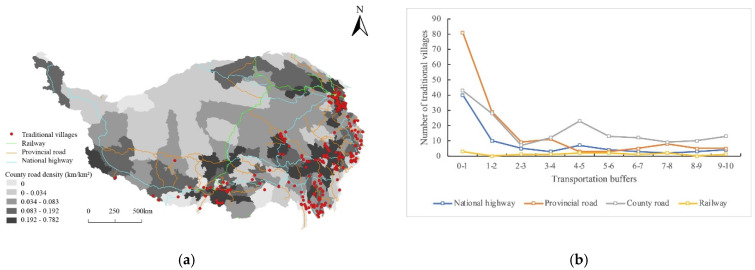
Distribution of traditional villages coupled with transportation systems on the Tibetan Plateau: (**a**) the spatial overlay analysis of village distribution and transportation systems; (**b**) the statistics of traditional villages in different transportation buffers.

**Figure 14 ijerph-19-13170-f014:**
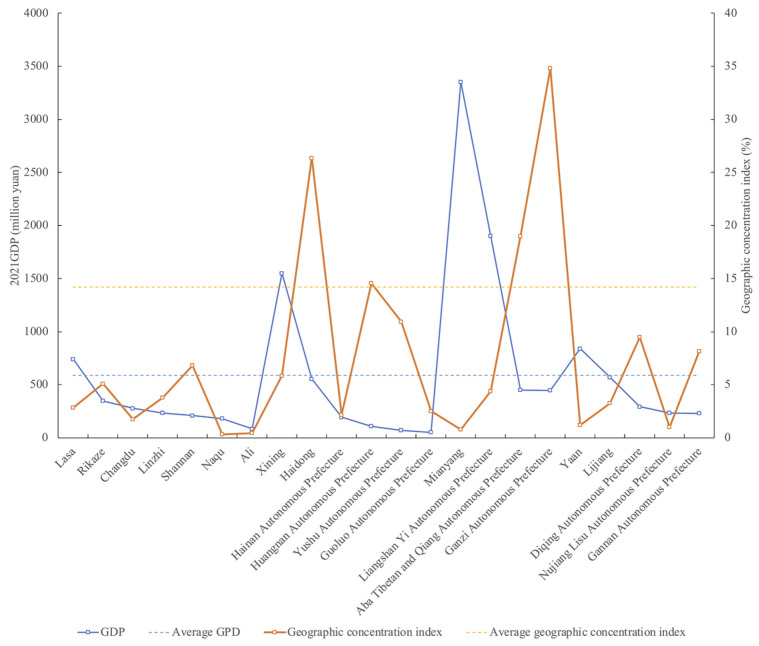
Economic development level and traditional village distribution on the Tibetan Plateau.

**Table 1 ijerph-19-13170-t001:** Data sources.

Project	Area	Number	Sources
Geographical data	Tibetan Plateau	n/a	Tibetan Plateau National Data Center of China: http://www.tpdc.ac.cn/zh-hans/ (accessed on 12 May 2022);
Socio-economic data	Tibetan Plateau	n/a	Tibetan Plateau National Data Center of China: http://www.tpdc.ac.cn/zh-hans/ (accessed on 5 March 2022); China National Bureau of Statistics: http://www.stats.gov.cn/ (accessed on 16 March 2022)
Maps	World and China	n/a	The Ministry of Natural Resources of the P.R.C.: http://bzdt.ch.mnr.gov.cn/ (accessed on 29 July 2022)
Traditional villages	Point data	312	Traditional Chinese villages digital museum: http://www.dmctv.cn/ (accessed on 5 March 2022); Ministry of Housing and Urban-Rural Development of the P.R.C.: http://www.mohurd.gov.cn/ (accessed on 8 March 2022)

**Table 2 ijerph-19-13170-t002:** Analytical models and interpretations.

No.	Index	Model	Model Definition	Geographical Interpretation
1	Nearest neighbor index	$R=\bar{D}∕{\bar{D}}_{i}$	*R* is the nearest neighbor index; $\bar{D}$ is the actual nearest neighbor distance; ${\bar{D}}_{i}$ is the theoretical nearest neighbor distance.	Reflects the spatial distribution type of point elements. When *R* > 1, it is uniformly distributed; when *R* = 1, it is randomly distributed; when *R* < 1, it is a clustered distribution.
2	Kernel density	$f(x)=\frac{1}{nh}\overset{n}{\underset{i=1}{\sum }}k(\frac{\chi -\chi_{i}}{h})$	f(x) represents the kernel density estimation of the kernel density function; n represents the number of points in the neighborhood; h represents the bandwidth; $k(\frac{\chi -\chi_{i}}{h})$ represents the kernel function.	Reflects the discrete degree of point features; the larger the f(x) value, the denser the distribution of point features.
3	Imbalance index	$S=\frac{\sum_{i=1}^{n}Y_{i}-50\left(n+1\right)}{100\times n-50\left(n+1\right)}$	n is the number of study areas (provinces); $Y_{i}$ is the cumulative proportion of the ith rank after the proportion of point elements in each region is ranked from large to small	Reflects the imbalanced degree of point elements in the area; 0 < *S* < 1 indicates uneven distribution, *S* = 1 indicates uniform distribution, and *S* = 0 indicates highly concentrated distribution.
4	Geographic concentration index	$G=100\times \sqrt{\overset{n}{\underset{i=1}{\sum }}{\left(\frac{\chi_{i}}{T}\right)}^{2}}$	G represents the geographic concentration index; $\chi_{i}$ is the number of point elements in the ith city area; *T* is the total number of point elements; n is the total number of cities	Reflects the concentration of point elements in a certain area; the value of *G* is between 0 and 100. The larger the value of *G*, the more concentrated the distribution of point elements.
5	Global Moran’s I	Moran’s I $=\frac{\sum_{i=1}^{n}\sum_{j=1}^{n}w_{ij}\left(x_{i}-\bar{x}\right)\left(x_{j}-\bar{x}\right)}{\sum_{i=1}^{n}\sum_{j=1}^{n}w_{ij}\sum_{i=1}^{n}{\left(x_{i}-\bar{x}\right)}^{2}}$	$x_{i}$ and $x_{j}$ are the number of villages in No.i county and No.j county, $\bar{x}$ is the average number of villages; $w_{ij}$ is the spatial adjacent weight matrix of the counties, n is the total number of counties.	The values of Moran’s I range from −1 to +1, the value of +1 meaning strong positive spatial autocorrelation, to 0 meaning a random pattern, and to −1 indicating strong negative spatial autocorrelation.
6	Pearson’s correlation analysis	${}_{\rho_{x,y}=\frac{N\sum XY-N\sum XY}{\sqrt{N\sum x^{2}-{\left(\sum X\right)}^{2}}\sqrt{N\sum y^{2}-{\left(\sum y\right)}^{2}}}}$ $t=\rho \sqrt{\frac{n-2}{1-\rho^{2}}}$	$\rho_{x,y}$ represents the Pearson’s correlation coefficient; *X* and *Y* refers to two variables; *t* quantifies the statistical significance	Reflects the correlation strength between two variables and its value varies from −1 to 1; the closer the value is to −1 or 1, the stronger the correlation might be; the closer to 0, the less significant of the correlation.

**Table 3 ijerph-19-13170-t003:** Distribution of traditional villages in provinces on the Tibetan Plateau.

Rank	Province	Total Number of Villages	Proportion	Cumulative- Proportion	Average Number of Villages (Per 10,000 km^2^)
1	Qinghai Province	121	39.78%	39.78%	1.66
2	Sichuan Province	120	38.46%	77.24%	4.76
3	Tibet Autonomous Region	34	10.90%	88.14%	0.28
4	Yunnan Province	33	10.58%	95.51%	8.55
5	Gansu Province	14	4.49%	100%	3.63
6	Xinjiang Uygur Autonomous Region	0	0	100%	0

**Table 4 ijerph-19-13170-t004:** Statistics related to traditional villages at the city level on the Tibetan Plateau.

Rank	Province/Region	City/Prefecture	Coverage of the Tibetan Plateau	Number of Villages	Proportion	Geographic Concentration Index
1	Sichuan	Ganzi Autonomous Prefecture	Almost whole	71	23.36%	34.81
2	Qinghai	Haidong	whole	57	18.75%	26.35
3	Sichuan	Aba Tibetan and Qiang Autonomous Prefecture	Almost whole	38	9.87%	18.98
4	Qinghai	Huangnan Autonomous Prefecture	whole	30	9.87%	14.56
5	Qinghai	Yushu Autonomous Prefecture	whole	17	5.59%	10.94
6	Yunnan	Diqing Autonomous Prefecture	part	15	4.93%	9.48
7	Gansu	Gannan Autonomous Prefecture	whole	14	4.61%	8.17
8	Tibet Autonomous Region	Shannan	whole	11	3.62%	6.83
9	Qinghai	Xining	part	9	2.96%	5.85
10	Tibet Autonomous Region	Rikaze	whole	8	2.63%	5.09
11	Sichuan	Liangshan Yi Autonomous Prefecture	part	7	2.30%	4.39
12	Yunnan	Lijiang	whole	6	1.97%	3.78
13	Tibet Autonomous Region	Linzhi	part	5	1.64%	3.25
14	Tibet Autonomous Region	Changdu	whole	4	1.32%	2.83
15	Tibet Autonomous Region	Lasa	whole	4	1.32%	2.52
16	Qinghai	Guoluo Autonomous Prefecture	whole	4	1.32%	2.17
17	Qinghai	Hainan Autonomous Prefecture	whole	4	1.32%	1.76
18	Yunnan	Nujiang Lisu Autonomous Prefecture	part	2	0.66%	1.20
19	Sichuan	Mianyang	part	2	0.66%	1.01
20	Sichuan	Yaan	part	2	0.66%	0.79
21	Tibet Autonomous Region	Ngari	whole	1	0.33%	0.45
22	Tibet Autonomous Region	Naqu	whole	1	0.33%	0.32

**Table 5 ijerph-19-13170-t005:** Ethnic make-up of traditional villages on the Tibetan Plateau.

Rank	Ethnic Group	Number of Villages	Proportion of Total
1	Tibetan	220	70.51%
2	Multi-ethnic mix	27	8.65%
3	Sala	17	5.45%
4	Han	12	3.85%
5	Qiang	11	3.53%
6	Tu	11	3.53%
7	Nakhi	6	1.92%
8	Lisu	3	0.96%
9	Hui	2	0.64%
10	Yi	1	0.32%
11	Nu	1	0.32%
12	Mosuo	1	0.32%
Total	n/a	312	100%

## Data Availability

Not applicable.

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
