# Peer review of "Spatial Distribution Characteristics and Influencing Factors of Traditional Villages on the Tibetan Plateau in China"

_ijerph, 2022, doi:10.3390/ijerph192013170_

Round 1

Reviewer 1 Report

1. In this paper, the overall spatial distribution pattern of traditional villages on the Qinghai-Tibet Plateau was studied. The scale of the study was large and only four indicators were used to describe its characteristics, resulting in a lack of research depth. Moreover, the author determined six spatial distribution patterns only by the kernel density index, which led to the lack of rigor in the division of them.

2. Why should such indices as Nearest neighbor index, Kernel density, Imbalance Index and Geographic Concentration Index be used to characterize the spatial distribution characteristics of traditional villages? Are these four indicators repeatable in describing the distribution characteristics of rural settlements, for example, kernel density index may be able to reflect the concentration degree of rural settlement distribution instead of the imbalance level? Can only these four indicators be used to describe the distribution characteristics of traditional villages reasonably and adequately?

3. In the study at the provincial level, it is not typical to compare the distribution differences of traditional villages in each province with the number of villages. For example, when the number of villages is similar, the difference of provincial area would lead to significant differences in the distribution of villages, so the the index taking into account the size of the site may be more convincing.

4. In the study of influencing factors of spatial distribution of traditional villages, the author only describes the distribution characteristics of villages based on different factors, but does not explain the degree and mechanism of them. It is suggested to use regression analysis and other methods to deepen the study in this part.

5. In the discussion, the author puts forward protection measures for the traditional villages of the Tibetan Plateau. However, these measures are not closely related to the conclusions of the study. It is suggested to put forward specific and actionable suggestions based on the research content and conclusions.

6. The Tibetan Plateau is mainly inhabited by nomadic people, which is very different from the farming people. It may be unreasonable for the author to choose the method of studying the rural settlement space in the farming ethnic areas to study the nomadic ethnic areas.

7. The author chose the special region of Qinghai-Tibet Plateau as the study area, but did not combine the special factors of climate, topography and nationality to conduct in-depth research, which happened to lose the characteristics of the original topic. It is suggested to find reasonable and quantifiable indicators with regional characteristics in the aspects of nature, culture, society and economy to highlight the site characteristics of the Qinghai-Tibet Plateau.

8.There is no breakthrough in the method and route of this study, which is still stuck in the traditional contents in the field of rural settlement morphology research a few years ago, so how to reflect the innovation of this paper?

Reviewer 2 Report

Thank you for submitting your work “Spatial Distribution Characteristics and Influencing Factors of Traditional Villages on Tibetan Plateau” to the International Journal of Environmental Research and Public Health

The paper draws attention to a relevant subject. It is very interesting and well organized. However, several issues need to be addressed properly before the paper is considered for publication.

My comments including major and minor concerns are given below.

I suggest reorganizing the abstract, highlighting the novelties introduced and the main numerical results. It should contain answers to the following questions:

Ø  What problem was studied and why is it important?

Ø  What methods were used?

Ø  What conclusions can be drawn from the results?

Ø  What is the novelty of the work and where does it go beyond previous efforts in the literature?

A very important aspect is related to the bibliography. I suggest enriching the state of the art. It might be useful to add a paragraph regarding the literature review, within which to highlight the innovative aspects of this study, compared to already published work on the same topic.

At the end of the introduction add the innovation of this study compared with the others present in the literature.

The methodology presentation section should be implemented. For example, a flow chart of the followed methodology can be added.

In table 1, the sources can be entered as numeric references and reported in the bibliography list. In addition, I suggest summarizing the contents of Table 1.

Please add a short description between two consecutive paragraphs.

Although Figure 1 shows an enlargement of a geographical area, it is not well visible. I suggest increasing the quality of the image.

In figure 2, the characters and lines of the legend are not easily readable.

The reference to the climate classification of the locality could be helpful, for example, the Koppen climate classification could be used.

More comments should be added to all graphs.

Image captions could be more detailed in order to make them understandable.

Figures 10 and 11 could be compacted (also Figures 13 and, Figures 15 and 16), and become one figure with codes a and b.

The caption of figure 12 should contain the description of figures a and b.

Figure 17 is not very visible.

Different fonts were used within the text.

Please add the table of Nomenclature with the explanation of all parameters present in the text.

The innovation of the study should be implemented in the conclusions. In addition, conclusions could report not only the study's innovation but also weaknesses and potential future developments.

Round 2

Reviewer 1 Report

1.The abstract needs to be more rigorous in its presentation. In the abstract, the author believes that economy and tourism cause the destruction of traditional villages, but the main text does not combine these two aspects to carry out the study.

2. The author mainly studies traditional villages, but does not define the concept of traditional villages in the paper. In addition, not all existing villages are developed from traditional villages, and the author uses existing villages to refer to traditional villages, which is unscientific.

3. 3.1.1 and 3.1.4 of the manuscript describe the distribution characteristics of villages on a wide scale. Why not write them in one chapter instead of two? In addition, the author spends a lot of space describing the distribution characteristics of villages with different indicators, which is similar to the accumulation of evaluation results of different indicators, but the core is actually described from the dimensions of whole region, province, city and nation. It is suggested to reorganize the logical structure of Chapter 3 to make it clearer and clearer.

4.When analyzing the spatial distribution characteristics of villages (3.1.2), why should the Imbalance index be used to characterize the distribution of villages at the provincial level and the Geographical concentration index be used to characterize the distribution of villages at the city level? What are the reasons for choosing these two indexes? Don't you use the same exponent?

5. How to obtain the ethnic type data of traditional villages? Please indicate in the manuscript.

6. The author mainly describes the spatial distribution characteristics of traditional villages on the Tibetan Plateau, and discusses the distribution characteristics of rural settlements based on possible influencing factors, without revealing how these influencing factors affect the spatial distribution of traditional villages, and what the degree of interaction between different influencing factors is. Further research is needed on the influencing mechanism of the spatial distribution of traditional villages on the Tibetan Plateau.

Reviewer 2 Report

All revisions have been made.
